# The CMEMS GlobColour Chlorophyll-a Product Based on Satellite Observation: multi-sensor merging and flagging strategies.

Philippe Garnesson[1], Antoine Mangin[1], Odile Fanton d'Andon[1], Julien Demaria[1], Marine Bretagnon[1]

[1]ACRI-ST, Sophia-Antipolis, 06904, France

*Correspondence to:* Philippe Garnesson (Philippe.garnesson@acri-st.fr)

**Abstract.** This paper concerns the GlobColour merged chlorophyll-a products based on Satellite Observation (SeaWiFS, MERIS, MODIS, VIIRS and OLCI) and disseminated in the framework of the Copernicus Marine Environmental Monitoring Service (CMEMS).

This work highlights the main advantages provided by the Copernicus GlobColour processor which is used to serve CMEMS

with a long time series from 1997 to present at Global level (4 km of spatial resolution) and for the Atlantic level 4 product (1 km).

To compute the merged chlorophyll-a product, two major topics are discussed:

- The strategy for merging remote sensing data; for which two options are considered. On the one hand, a merged chlorophyll-a product computed from a prior merging of the remote-sensing reflectance of a set of sensors. On another

hand, a merged chlorophyll-a product resulting from a combination of chlorophyll-a products computed for each sensor.

- The flagging strategy used to discard non-significant observations (e.g. clouds, high glint…)

These topics are illustrated by comparing the CMEMS GlobColour products provided by ACRI-ST (Garnesson et al., 2019) with the OC-CCI/C3S project (Sathyendranath et al., 2018). While GlobColour merges chlorophyll-a products with a specific flagging, the OC-CCI approach is based on a prior reflectance merging before chlorophyll-a derivation and uses a more

constrained flagging approach.

Although this work addresses these two topics, it does not pretend to provide a full comparison of the two datasets, which will require a better characterisation and additional inter-comparison with in situ data.

## 1 Introduction

The Copernicus Marine Environmental Service (CMEMS) provides regular and systematic reference information on the physical state and on marine ecosystems for global oceans and for European regional seas (temperature, currents, salinity, sea surface height, sea ice, marine optical properties, etc.).

This capacity encompasses satellite and in situ data-derived products, the description of the current situation (analysis), the prediction of the situation a few days ahead (forecast), and the provision of consistent retrospective data records for recent years (re-analysis).

The Ocean Thematic Assembly Centre (OCTAC) is part of CMEMS and is dedicated to the dissemination of Ocean Colour (OC) products derived from Satellite based remote sensing (Le Traon et al., 2015). OCTAC provides Global and Regional (Arctic, Atlantic, Baltic, Black Sea and Mediterranean) products for the period spanning from 1997 to present.

For Global products, the Copernicus GlobColour processor is used operationally since 2009 to serve CMEMS and its precursors (a series of EU research projects called MyOcean).

The GlobColour processor has initially been developed in the framework of the GlobColour project started in 2005 as an ESA Data User Element (DUE) project to provide a continuous data set of merged L3 Ocean Colour products. Since the beginning of the project, GlobColour has been continuously used by more than 600 users worldwide. This effort has been continued in the framework of CMEMS to derive (among others) the chlorophyll-a Ocean Colour core product.

Many algorithms have been published to retrieve chlorophyll-a from reflectance data derived from satellite observations (e.g., Muller Karger et al., 1990, Aiken et al., 1995, Morel 1997, O'Reilly et al., 1998, 2000). Since early 2018, the CMEMS GlobColour chlorophyll-a product is based on the merging of 3 algorithms:

- the CI approach (Hu, 2012) for oligotrophic water (70% to 80% of ocean),
- the common approach OCx (OC3, OC4, OC4Me depending on the sensor) for mesotrophic water
- the OC5 algorithm (Gohin et al., 2002) for complex waters, which is of specific interest for end users who manage complex waters along the coastal zone.

It should be noted that the OC5 algorithm is based on a lookup table implementation which handles both complex and mesotrophic waters (see section 2.1 Merging approach)

The work presented here highlights the conceptual advantage of the CMEMS Copernicus GlobColour processor with regards to flagging and merging of sensors. In the following sections, results are described and illustrated with comparison to the OC-CCI products.

The comparison between GlobColour and OC-CCI is especially relevant since the same chlorophyll-a algorithms (CI and OC5) are used by the two initiatives: it implies the differences in performance mainly result from the merging strategies and flagging schemes.

## 2 Methods

### 2.1 Merging approach

The CMEMS GlobColour merged chlorophyll-a product relies at present on the following sensors: SeaWiFS (1997-2010), MERIS (2002-2012), MODIS-Aqua (2002-present), VIIRS-NPP (2012-present) and OLCI-S3A (2016-present).

The long time series from 1997 to present relies on different sensors, observing the Earth at different spectral bands (and different bandwidth), with different acquisition times (so different atmospheric and sun conditions), and with different spatial resolutions from about 300 meters to 1 km at nadir (larger on the swath border). Main characteristics of the sensors/bands used for CMEMS are summarized in Table 1. VIIRS-NOOA20 and OLCI-S3B will be ingested in the operational products in 2019. It should be noted that the global chlorophyll-a product is at present provided at a 4km spatial resolution, but the objective in

the coming years is to provide, at least along the coast, a chlorophyll-a product at 300 meters resolution.

All sensors used observe the Earth along a helio-synchronous orbit. Figure 1 displays the coverage of a single sensor which is unable to provide the full Earth coverage over a single day (Maritorena et al., 2010). VIIRS provides a larger swath than the other sensors, but the coverage is incomplete because of sun glint.

When more than one sensor is available for the same period it is possible to take advantage of their complementarity and

redundancy for a number of benefits. For instance, since they may record the same spot at different times of the day, morning haze can impact one sensor and not another (Toole et al., 2000).

To compute a multi-sensor chlorophyll-a product, mainly two merging approaches exist.

- A first approach (used by OC-CCI) is based on merged Reflectance of Remote-Sensing (RRS) computed in a prior step and then used to derive chlorophyll-a. Based on a band shifting and bias correction approach, merged RRS for the standard

SeaWiFS wavelengths (412, 443, 490, 510, 555 and 670 nm) are provided. The blended chlorophyll-a algorithm used in the OC-CCI v3.1 release attempts to weight the outputs of the best performing algorithms based on the water types in presence.

- Conversely, in a second approach (used by CMEMS GlobColour), chlorophyll-a is computed in an initialisation step for each sensor using its specific characteristics (spectral band, resolution), and then the mono-sensor chlorophyll-a products

are resampled and merged. The continuity of the different algorithms was initially obtained using a water classification approach (Saulquin al., 2018). Starting in early 2018, a new approach has been adopted:

  o First, the continuity of algorithms used for mesotrophic and complex waters are guaranteed by the OC5 lookup table. The OC5 lookup table is initialized using the OC3 and OC4 coefficients from agencies and then empirically adjusted when the green band exceeds a given threshold (see Gohin et al., 2002 for details).

o Then, the CI and OC5 continuity is ensured using the same approach as NASA. When the chlorophyll-a concentration is in the range 0.15 to 0.2 mg.m$^{-3}$, a linear interpolation of OC5 and CI is used. This provides continuity between the two algorithms.

## 2.2 Flagging approach

Inputs of the Copernicus GlobColour processor are the level 2 products provided by the space agencies. To derive chlorophyll-a, the input level 2 reflectance and input flags are used (level 2 provides flags/indicators about the quality of the reflectance at pixel basis). For instance, a pixel can be impacted by a sun glint effect. In such case reflectance data is available, but it is recommended by agencies not to use it for further processing. Each space agency publishes flagging data based on its own strategy which has been designed to guarantee the quality of its products. The drawback is that for complex water (especially along coastal area), a large proportion of the data is flagged, resulting in level 3 products with limited coverage.

Since early 2018, the GlobColour processor has been modified to apply a lower level of flagging resulting in a better spatial and temporal coverage. This modification is inherited from the OC5 algorithm (Gohin et al., 2002) which was initially designed for coastal monitoring. OC5 uses its own strategy to flag data: the algorithm uses both the official flags and empirical thresholds that have been tuned for each sensor (e .g. the OC5 sun zenith angle (SZA) is set to 78° instead 70°).

The OC5 flagging strategy is used by the GlobColour sensor approach but not by the OC-CCI approach which uses a specific OC5 lookup table applied on the merged reflectances.

Concerning OC-CCI, the flagging strategy for release v3.1 depends on the sensor. When the reflectance data originates from level 2 data provided by the agency (SeaWiFS and VIIRS-SNPP) the official flags are applied. When the POLYMER algorithm (Steinmetz et al. 2011) is used for atmospheric correction (MERIS and MODIS) a generic pixel identification and classification algorithm called Idepix is used (part of BEAM software) instead the standard POLYMER flagging (which is too permissive).

## 3 Results and Discussion

### 3.1 The reflectance merging approach

The reflectance merging approach is used by OC-CCI to derive the global Chlorophyll-a OC-CCI timeseries and by the regional CMEMS products. As pointed by Volpe et al. (2019) for the regional Mediterranean products, the band merging approach has the advantage of providing a homogeneous dataset of spectral reflectance from which can be derived, in full consistency for the long term, different environmental parameters, among them chlorophyll-a products but also light attenuation, Kd, Suspended Particulate Matter and others.

However, the consistency of the long-time series provided by OC-CCI suffers from some limitations. Figure 2, from OC-OC-CCI Product user guide, intercompares the different releases of the OC-CCI time series. It shows that the V2 release was strongly impacted by the MERIS sensor stopping operations in April 2012 (see Table 1). The V3 release demonstrates a trend depending on the sensors used: It increases for the period 2002-2010 (based on the contributions of SeaWiFS, MODIS and MERIS and decrease for the period 20122017 as only MODIS and VIIRS-SNPP are used.

On a regional scale, strong limitations on the consistency of the time-series and trends derived are observed as illustrated for the Arctic on Figure 3. Strong variations are also observed throughout the years. For instance, before 2002, only SeaWiFS is available, limiting the quality of the output data. Then the end of the MERIS lifetime and start of VIIRS SNPP in 2012 causes the same trend as described above.

It is known that both MODIS and VIIRS instruments have major calibration issues starting in about 2012. VIIRS-SNPP degradation has been identified a few months after launch and MODIS, while designed for a lifetime of 7 years, is still operating after 17 years. MODIS calibration has required regular modifications to adjust the temporal trends (R2009.1, R2010.0 and especially R2012.0). At the end of 2017, a NASA reprocessing called R2018 has significantly alleviated VIIRS issues and more importantly corrected the MODIS drift with a new procedure to regularly update the MODIS calibration (available about 10   3 months after acquisition).

It should be noted that this new NASA processing (called R2018.1) does not yet benefit the full OC-CCI series (only for the recent OC-CCI v3.1 extension until June 2018) but a new OC-CCI v4 release is scheduled for 2019.

It should also be noted that the R2018 dataset still suffers from issues which will continue to impact future reprocessing 15   attempts. Indeed, VIIRS RRS443 and RRS488 increase regularly since 2012 while MODIS is comparatively more stable.

Figure **4** shows the relative RRS443 difference between VIIRS and MODIS (in %) based on the monthly NASA R2018 global products at 4km. VIIRS suffers from a significant drift since its launch as illustrated with January evolution for years 2012, 2016, 2019. In January 2012, 90 % of MODIS pixels at global level were higher than VIIRS-SNPP, while in January 2019 100% of VIIRS-SNPP pixels were higher than MODIS.

Another major difficulty for merging RRS for the different sensors is that the observed bias varies according to the region and season considered as previously shown with artificial trends along the years.

Figure **5** shows the inter-comparison of RRS at about 670nm between VIIRS-SNPP and OLCI-S3A compared to MODIS. It shows very important bias (e.g. 82% of the pixels of OLCI exceeds a relative difference of 20%), and in the case of MODIS, the equatorial zone has a different behaviour than high latitude.

Impact of such reflectance merging on chlorophyll-a is showed on the Figure 6. For the year 2018, it shows limitation in its ability to handle clear water (the colour scale has been set to the range 0.01 to 0.2 mg/l): discontinuities between tracks of the sensor clearly appear.

The previous illustrations demonstrate the limitation of the assumption of consistency along the OC-CCI time series (and at daily basis on Figure 6) but it is clear that the GlobColour products are also impacted by the quality of input RRS upstream.

Figure 7 shows the month to month evolution of the chlorophyll-a median concentration computed at global level for OC-CCI and GlobColour products. To make the two datasets inter-comparable, monthly statistics are computed on common pixels. The yellow sections on the plot shows the change of sensor combinations. Trends and gaps should be carefully interpreted while taking into account the sensor used. The oscillations of the median through time are linked to the sensor coverage which is moving from North to South for high latitude (due to the variation of the sun zenith angle which render measurements in winter

unfeasible). At the beginning of SeaWiFS (Sept 1997 to mid-1998) data was only partially available, resulting in limited coverage. When MODIS and MERIS started in 2002 spatial coverage was improved. It means that the gap at this date does not correspond to a change in the continuity of the time-series but a change in the spatial coverage of pixels used to compute the median. The higher values for OC-CCI are most likely linked to the NASA R2018 reprocessing which is used by GlobColour but not yet by OC-CCI. During this R2018 reprocessing, MOBY (in situ data) and SeaWiFS calibration were also improved with a resulting decrease in chlorophyll-a of order 10% for all sensors.

## 3.2 The chlorophyll-a merging approach

This approach is the one used by the GlobColour processor for the CMEMS products and is not designed to solve the issues linked to the upstream data. However, the sensor approach (instead starting from merged RRS) has crucial advantages compared to the previous one:

- When a new sensor (or a new reprocessing of an existing sensor) becomes available, limited efforts are required because the bias correction is limited to the chlorophyll-a field of the considered sensor. For the merged reflectance approach, the bias correction of 5 reflectance measurements and interpolations is required to simulate the band 510 (this band is not available for VIIRS-SNPP, VIIRS-JPPS1 and MODIS (see Table 1)).
- For the CI algorithm, the GlobColour processor benefits from the efforts of the space agencies to adjust the coefficients accounting for the high variability of the band 670 (
- Figure **5**) for each sensor.
- It should be noted that the chlorophyll-a algorithm is applied on the level 2 sensor grid while in the merging approach it is applied on the common grid required to merge the reflectance. The use of sensor level 2 grid guarantees that the algorithm is applied on reflectance with consistent time observations. On the other hand, when reflectance is re-projected on a common grid, it results in mixing pixels observed with an observation shift that can raise to 4 hours (see Table 1) in the case of MODIS and VIIRS-JPSS1). This consideration is of minor significance in the case of the Global products with a spatial resolution of 4 km but will become more significant when product will be provided at a resolution of 300 meters.

## 3.3 The flagging approach

When compared to the official agencies' recommendations, the OC5 flagging strategy significantly improves the spatial coverage of the product especially for NASA sensors. In the framework of CMEMS we have estimated that at sensor level the coverage is increased by a factor of 3.2 for VIIRS-NPP, 2.6 for MODIS-a , 1.6 for MERIS, 2 for SeaWiFS, and 1.3 for OLCI-S3A.

Therefore, for the merged product, GlobColour chlorophyll-a coverage is improved by a factor 2.8 compared to OC-CCI. For the period 2002-2012 the increase in coverage is limited to a factor 1.5.

Figure **8** and

Figure **9** show the result of the flagging strategy depending the OC-CCI or GlobColour approach.

Figure 10 shows that the combination of the usage of this flagging strategy and OLCI leads to a considerable improvement of the coverage without creating additional artefacts. As of today, both products benefit of the latest NASA R2018 reprocessing.

Figure 11 shows that, in certain cases, the OC-CCI coverage could be better than the GlobColour one. However, in this example the OC-CCI approach is affected by significant noise, potentially due to cloud contamination. This noise might be due to level 2 inputs. Indeed, while GlobColour is using the level 2 from agencies, OC-CCI starts from level 1, applies POLYMER algorithm to MERIS and MODIS along with a specific flagging.

## 4 Conclusion

This work presents different ways to merge sensors and different flagging strategies to estimate the daily chlorophyll-a field.

Compared to the chlorophyll-a merging approach, the key advantage of the reflectance merging approach is that it provides a homogeneous dataset of spectral reflectance useful when deriving the chlorophyll-a product using a common algorithm. One could expect a better consistency for the long time series. However as illustrated above, this assumption is not true since the homogeneity of the spectral reflectance is at present not obtained (spatial and temporal discontinuities exist).

It should be underlined that this limitation also exists with the sensor chlorophyll-a merging approach: in both approaches if a temporal trend is observed it should be carefully analysed.

The present findings highlight the advantage of a Chlorophyll-a per sensor merging approach compared to the reflectance merging approach:

- The sensor approach facilitates the ingestion of a new sensor or a new reprocessing. Consequently, NASA R2018 and OLCI-S3A have been successfully introduced in April 2018 for the merged chlorophyll-a GlobColour chain but is not yet available in the other initiatives (OC-CCI or CMEMS regional product). The addition of VIIRS-NOOA20 (JPSS) and OLCI-S3B in the merged GlobColour chlorophyll-a product will occur in July 2019.

- It should be noted that the reflectance merging approach provides a limited set of six common spectral bands based on the SeaWiFS sensor. For more recent sensors (i.e. Modis, VIIRS-SNPP and VIIRS-JPPS1) only 5 native reflectance are available (see Table 1): the band at 510 nm is obtained by interpolation. Other extra bands from MERIS or MODIS or OLCI which are not part of the 6 bands are not usable in the reflectance merging approach (because the spatial complementarity of the sensors cannot be used at daily level and for the full time series). For the chlorophyll-a merging approach, when extra bands are available (i.e. MERIS, MODIS and OLCI) they can be used to improve the algorithm in the future. This perspective is already investigated to retrieve PFT from OLCI (e.g. Xi, 2018).

- The sensor approach provides an improved daily spatial coverage when OC5 is applied on the sensor reflectance (not on merged reflectance). For the period spanning from 2012 to present the spatial coverage is improved by an important factor (about 2.8) when compared to the OC-CCI product. Both open ocean and coastal areas are improved. It is required for many users involved in the EU Water Framework and Marine Strategy Framework Directive. To satisfy users interested in coastal data, a better spatial resolution (300 meters) is also required. From this point of view the chlorophyll-a merging approach is also more promising (algorithm can be applied on the level 2 track grid to limit the mixing of the pixels).

A better spatial coverage is also a key point to guarantee the quality of the CMEMS GlobColour chlorophyll-a "cloud free" product (called daily L4 in the CMEMS catalogue) which is based on a spatial and temporal interpolation of daily level 3 product. A better daily coverage limits the risk of artefact due to interpolation.

**Acknowledgement**

We thank Emmanuel Boss and another anonymous referee for their constructive comments which allowed us to improve the quality of this manuscript.

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

**Figure 1**: Swath of the different sensors used at present by CMEMS for (a) MODIS-Aqua, (b) VIIRS-NPP and (c) OLCI-S3A. In practice the effective swath coverage is reduced mainly due to clouds or sun glint effects.

Source : http:octac.acri.fr.

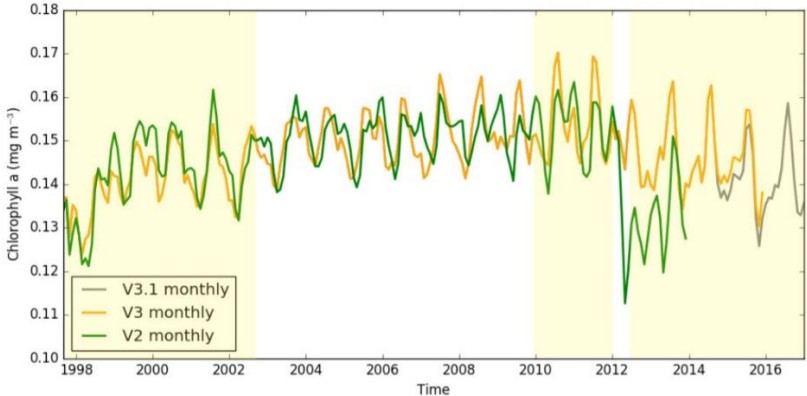

**Figure 2:** Comparison of the Global Median chlorophyll-a concentration as function of time for v2.0, v3.0 and v3.1 using the monthly composite as input. Source: OC-CCI Product User Guide, release 3.1.0, 24th of April 2017. Yellow part shows change of sensor combinations (see Table 1)

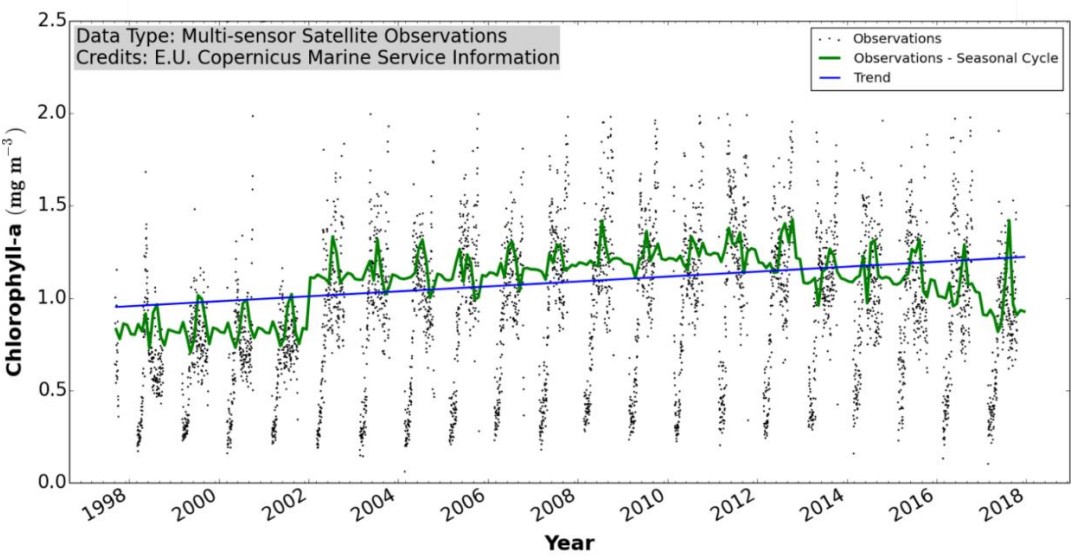

**Figure 3 :** Arctic time series and trend (1997-2017) from OC-CCI product. The time series are derived from the regional chlorophyll-a reprocessed (REP) products as distributed by CMEMS which, in turn, results from the application of the regional chlorophyll-a algorithms with remote sensing reflectances provided by the ESA Ocean Colour Climate Change Initiative (ESA OC-CCI). Daily regional mean values are calculated by performing the average (weighted by pixel area) over the region of interest. A fixed annual cycle is extracted from the original signal, using the Census-I method as described in Vantrepotte et al. (2009). The de-seasonalised time series is derived by subtracting the seasonal cycle from the original time series, and then fitted to a linear regression to obtain the linear trend. Source: CMEMS OMI QUID.

**Figure 4:** relative difference at RRS443 of VIIRS and MODIS (in %) based on the monthly NASA R2018 global products at 4km). VIIRS suffers from a significant trend since it has been launched as illustrated with January month evolution for years (a) 2012, (b) 2016 and (c) 2019.

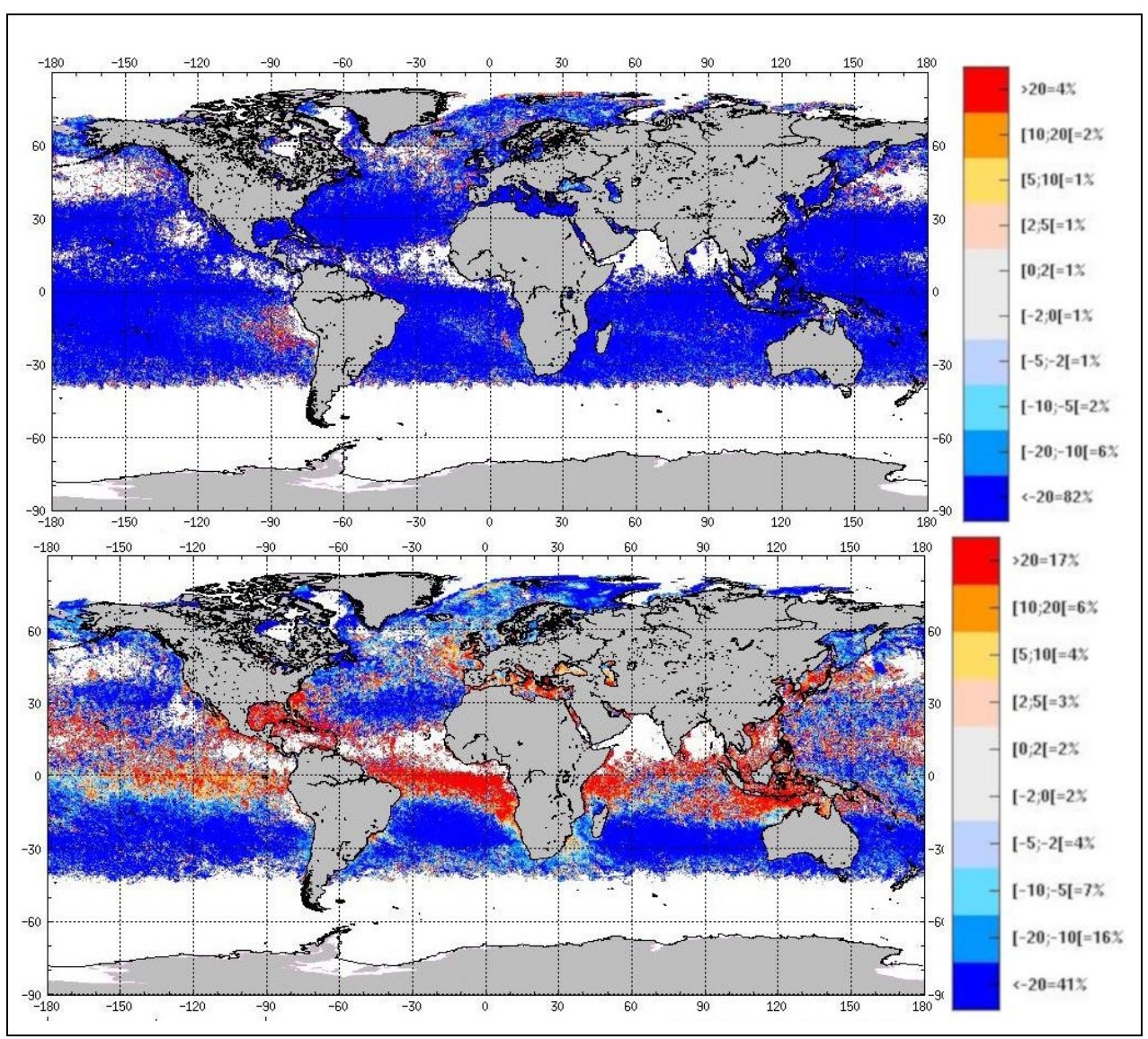

**Figure 5:** Relative difference between sensors [(S1-S2)/S2] of monthly RRS (June 2018), for (a) MODIS-RRS667 and OLCI-RRS665 and (b) VIIRS-RRS671 and MODIS-RRS667. Source: these plots are part of the monitoring done by the OCTAC and reported on http:octac.acri.fr.

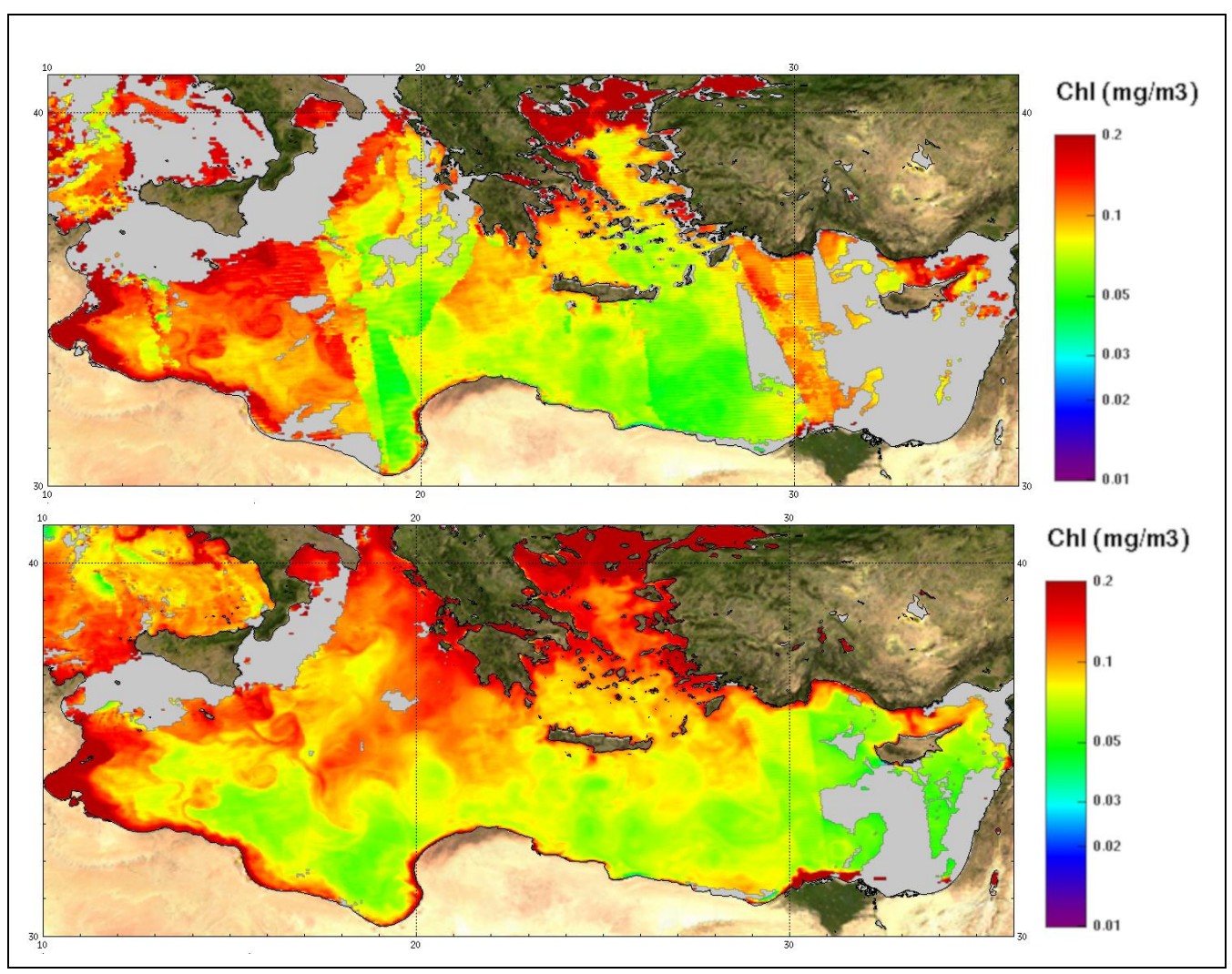

**Figure 6:** Inter-comparison of (a) OC-CCI and (b) GlobColour product (13 June 2018) for Oligotrophic water (CI algorithm). The color scale has been set to the range 0.01 to 0.2 mg/l: discontinuities between tracks of the sensor clearly appeared on the OC-CCI case at the top.

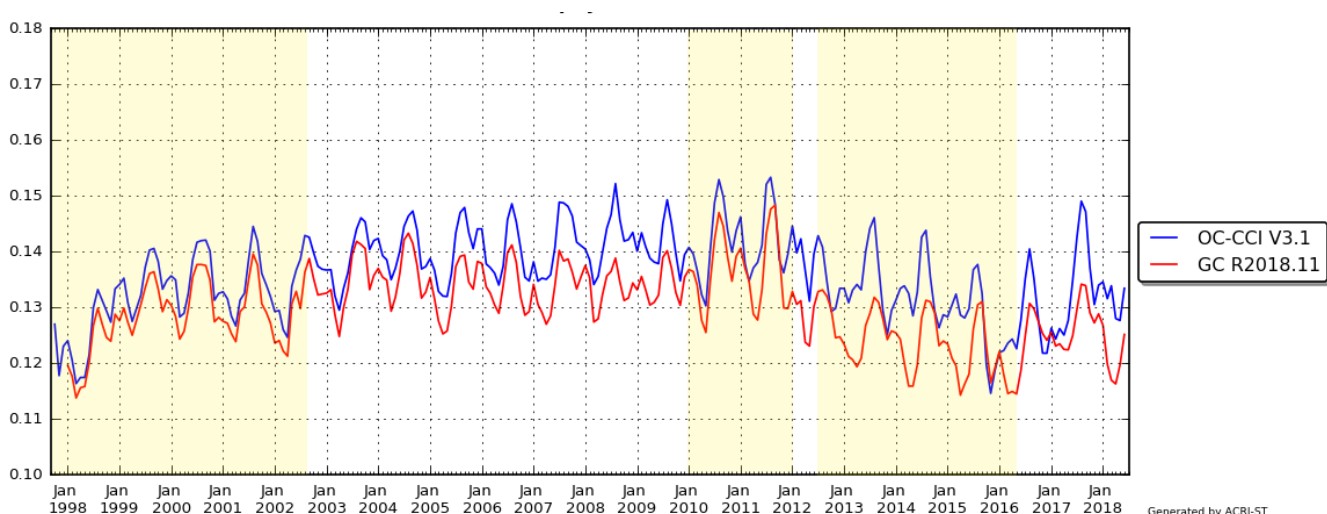

**Figure 7:** Comparison of the Global median chlorophyll-a concentration as function of time for OC-CCI v3.1 (blue line) and GlobColour R2018.11 (red line) at global level and for common pixels. Yellow part shows change of sensor combinations. The discontinuities and trends of the median along the time should be carefully interpreted according the sensors used.

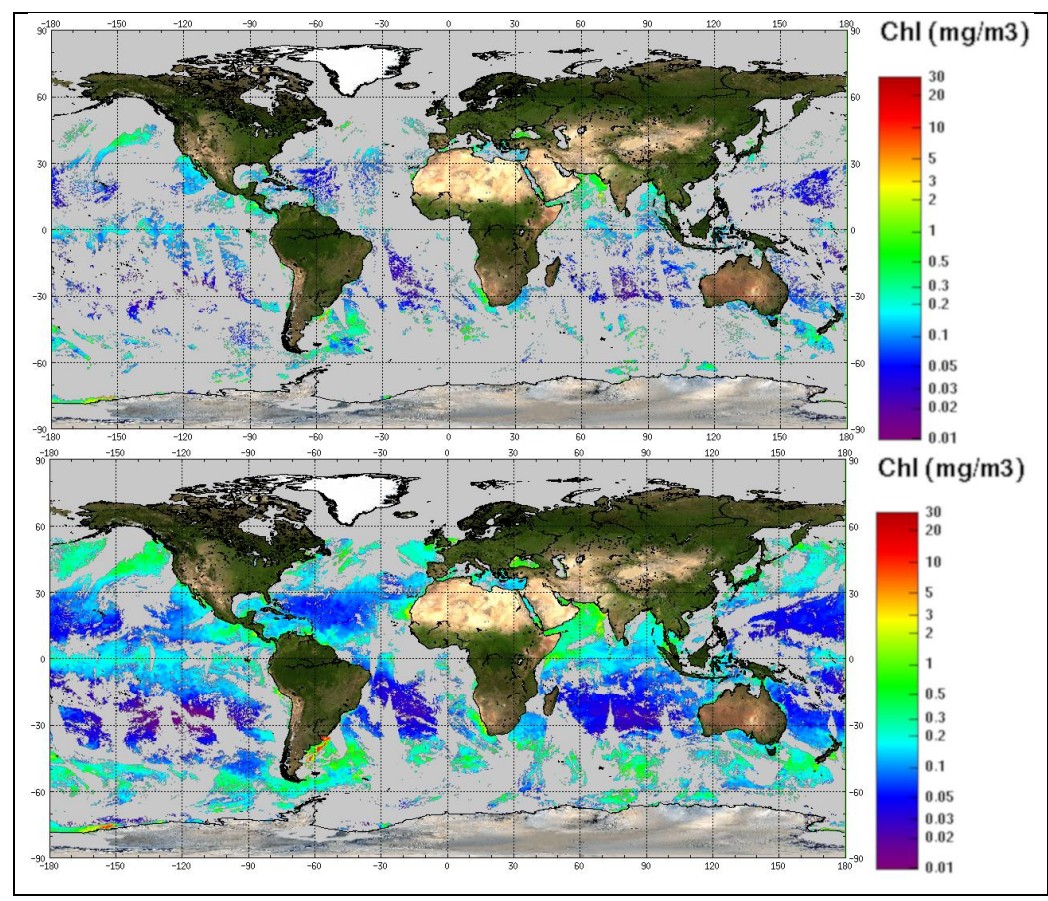

**Figure 8:** Chlorophyll-a concentration (15 Dec. 2017), (a) OC-CCI level 3 product and (b) GlobColour product. The two initiatives are using MODIS-A, and VIIRS-SNPP at this date, in complement OLCI-S3A is also used by GlobColour products

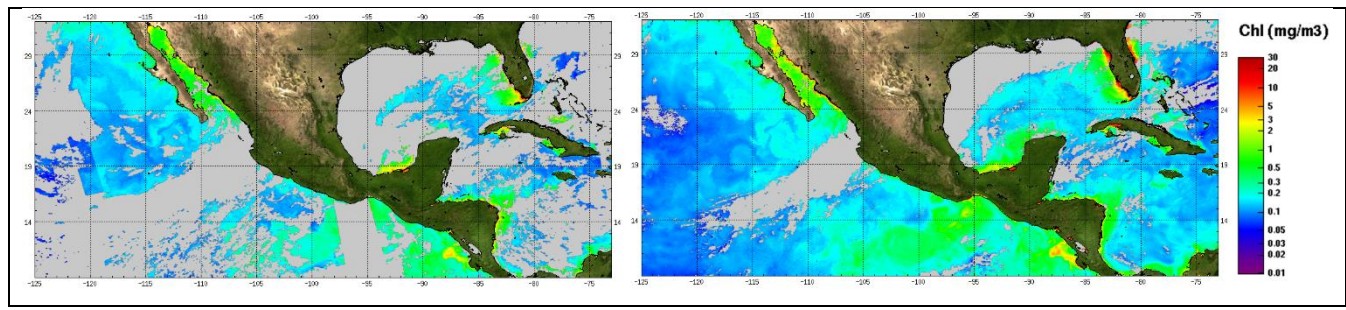

**Figure 9:** Chlorophyll-a concentration (15 Dec 2017): (a) OC-CCI level 3 product, (b) GlobColour product.

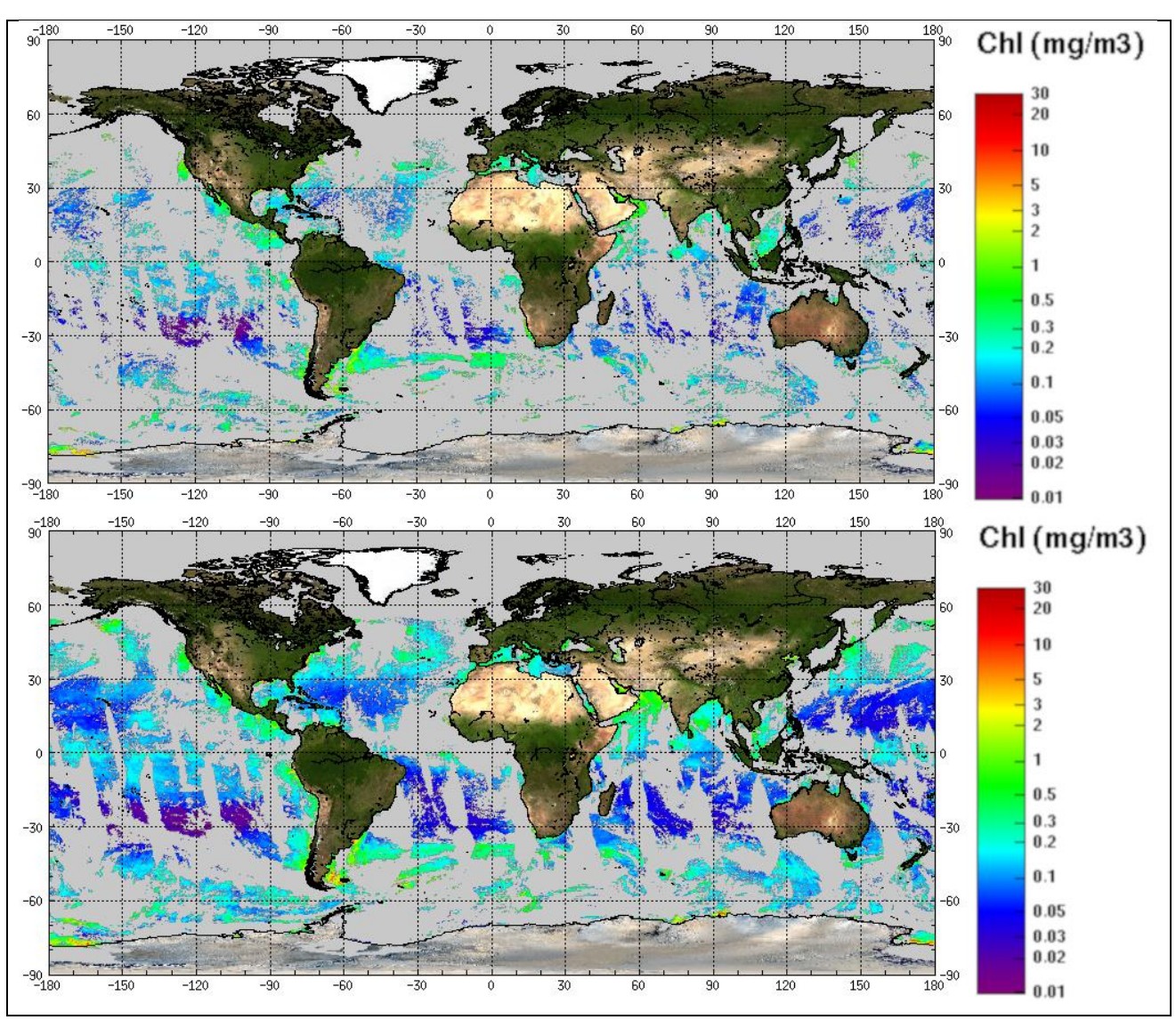

**Figure 10:** Chlorophyll-a concentration (1 Jan 2012), (a) OC-CCI level3 product and (b) GlobColour product.
The two initiatives are using MODIS-A, MERIS and VIIRS-SNPP at this date.

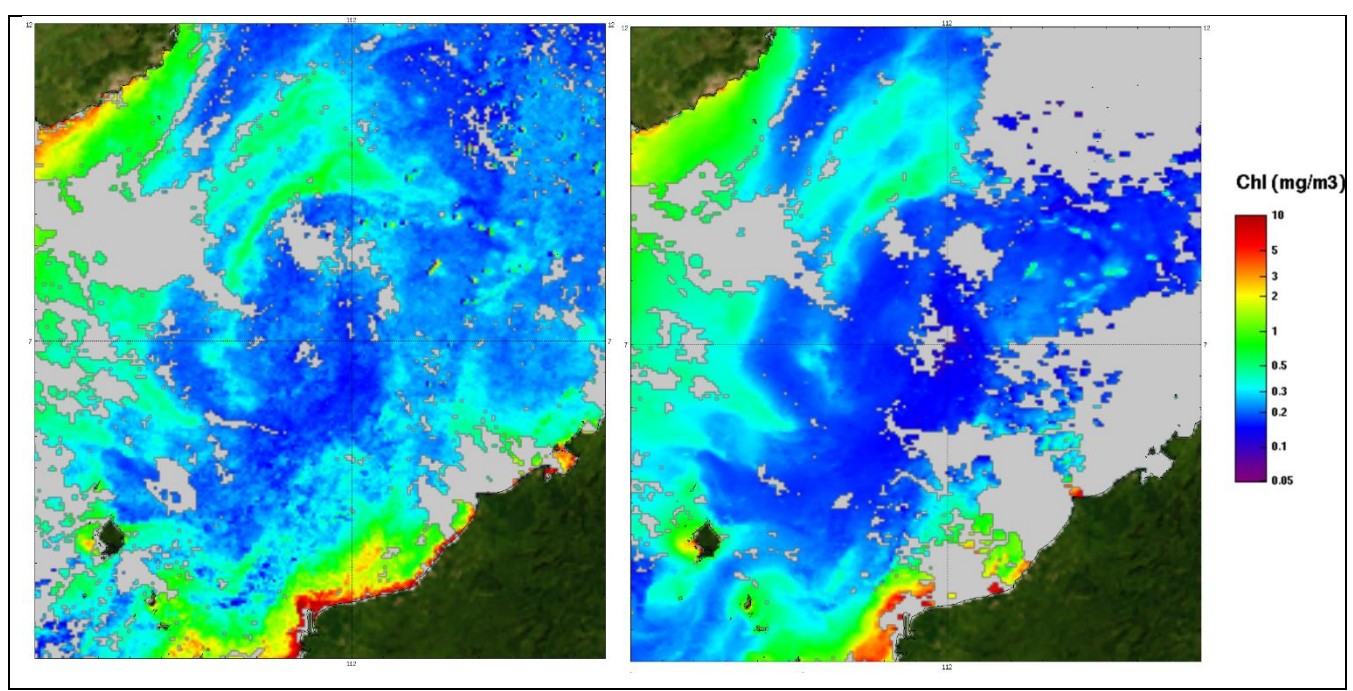

**Figure 11:** Chlorophyll-a concentration (1 Jan. 2012), (a) OC-CCI level3 product, (b) GlobColour product.

**Table 1:** Main characteristics of sensors/bands used for CMEMS (VIIRS-JPPS1 and OLCI-S3B will be used by the GlobColour processor in the framework of the CMEMS release scheduled in July 2019).

| Sensor | RRS Wavelengths (nm) | Spatial Resolution At Nadir (km) | Swath width (km) | Equator crossing time | Period |
|---|---|---|---|---|---|
| SeaWiFS | 412,443,490,510,555,670 | 1 & 4 | 1502 | 12:20 | 1997-2010 |
| MERIS | 413,443,490,510,560,620,667, 681,709 | 1 & 0.3 | 1150 | 10:00 | 2002-2012 |
| MODIS-Aqua | 412,443,469,488,531,547,555, 645,667,678 | 1 | 2330 | 13:30 | 2002-present |
| VIIRS-NPP | 410,443,486,551,671 | 1 | 3040 | 10:30 | 2012-present |
| OLCI S3A | 400,412,442,490,510,560,620, 665,674,681,709 | 1.2 & 0.3 | 1270 | 10:00 | 2016-present |
| VIIRS-JPPS1/NOAA20 | 411,445,489,556,667 | 0.75 | 3040 | 9:50 | Dec-2017-present |
| OLCI S3B | 400,412,442,490,510,560,620, 665,674,681,709 | 1.2 & 0.3 | 1270 | 10:00 | 2018-present |

