# Peer review of "The CMEMS GlobColour Chlorophyll-a Product Based on Satellite Observation: multi-sensor merging and flagging strategies."

_Ocean Science, 2018_

## Referee Comment (RC1) · Anonymous Referee #1 · 24 Jan 2019

This article presents the new version of the GlobColour product delivered by ACRI-ST within the CMEMS. As this GlobColour Chlorophyll-a (Chl-a) product has a global coverage and provides retrievals in coastal waters this manuscript can be of interest for many current and future users of satellite-derived products.

Chl-a in this new GlobColour product is derived from two algorithms: the Color Index of NASA for clear waters (Chl-a < 0.15 mg m-3) and the OC5 algorithm of Ifremer for water where Chl-a is superior to 0.2 mg m-3, including the coastal turbid waters. This is very similar to the strategy chosen by the Plymouth Marine Laboratory for the OC-CCI product, also provided at global scale. However a distinction is clearly made by the authors: the GlobColour processing chain provides a Level 3 Chl-a multi-sensor product obtained from mono-sensor Chl-a whilst the Level-3 Chl-a of OC-CCI is obtained from

a prior merging of the remote-sensing reflectance of the different sensors on a common reference of spectral bands (SeaWiFS). The OC-CCI approach is similar to that of the Mediterranean Product Unit of CMEMS described in Volpe et al. (Ocean science, accepted). Targeting directly Chl-a, the GlobColour processing can theoretically and practically be adapted more quickly to the modification of the products of any single sensor (following the reprocessing by the Agencies) whilst this task is more difficult to achieve through the complexity of the band switch and band correction operated in the OC-CCI approach. However as pointed out by Volpe et al., the band merging approach has the advantage of providing a homogeneous dataset of spectral reflectance from which can be derived, in full consistency for the long term, different environmental parameters, amongst them Chl-a but also light attenuation, Kd, Suspended Particulate Matter, . . ..

The authors discuss different issues encountered in the near real-time and long term processing of Ocean Colour data and some interesting illustrations are provided on the effect of the drift of Rrs in flight and the successive reprocessings by the Agencies onto the OC-CCI product ( Fig. 2 & 3) or onto the GSM/Nasa (Fig. 5.) product. However these major operational constraints have also consequences on the GlobColour and these latters are not described Thorough descriptions of the quality and flaws of the GlobColour products over the period 1997-2018 are missing. This could be due to the fact that this GlobColour product is new. In that case, would it not be better to change the title for something as "The new GlobColour product and the challenge of Ocean Colour processing at global scale?" The choices of the author in the article should be easier to understand.

In conclusion of these general remarks, the paper should be re-organized. For instance, the first paragraph of the Result and Discussion, 3.1, deals only with the OC-CCI and GSM products! Finally, in this text, the GlobColour product is assessed through its difference (more flexibility as for ingesting OLCI data) with other products but not through comparisons to in situ data; which is a main issue in OC applica-

tions, particularly in the coastal waters that are now clearly addressed by OC-CCI and GlobColour. This issue should, at minimum, appear in the discussion. As CMEMS OC products aim at covering coastal waters, why not take advantage of the coastal monitoring networks for a better flexibility in modifying the processing chains after a reprocessing by the Agencies or the availability of a new sensor?

Despite these comments, the manuscript is worth publishing. The illustrations are very informative and the main issues in the operational processing of OC data are pointed out.

Specific remarks

Introduction

Line 11 the CCI/Sv3 is mentioned but not referenced. That will be done later in the text.

Line 19 The continuity between OCx (OC3 & OC4) and OC5 is guaranteed by the construction of the OC5 tables. What do you mean, OC3 and OC4 are used in the GlobColour product in complement to CI and OC5?

2.2 Flagging approach This chapter is not clear. How is the OC5 LUT doing its own flagging? Not sure there is a control of the errors coming from the atmospheric correction or clouds by the ratios used in OC5. The 412 Rrs processed in OC5 can take into account a possible effect of the overcorrection of the atmospheric content but it could only be marginal in case or clouds, . . ..

3.1 Results are those of OC-CCI (already mentioned). However this chapter deals with interesting issues in OC monitoring.

In Figure 6, it would be better to show the 2 deviations with a same reference: MODIS Rrs 667. OLCI-MODIS and VIIRS-MODIS would appear with similar colours, demonstrating the variability of the atmospheric corrections between OLCI and VIIRS on one side and MODIS on the other.

Conclusion

It is not really a conclusion. The conclusion (or the last lines of the discussion) should open a window on a possible improvement of the processing chains; I would have appreciated a larger view, a strategy for improving the quality of the products, an opening to validation and flexibility after the launch of a new sensor or after a major reprocessing by the Agencies.

Technical corrections

Page 1:

Line 12: provides Line 24: the Ocean Colour Thematic Assembly Centre

Page 2:

Line 12; the CCI/S3v project is not defined Lines 16-19; the continuity of the 3 algorithms. CI , OC5 and ? Lines 20-21: the two sentences could be merged.

Page 3:

Lines 2 and 3. you already said (page 2 line 29) that VIIRS-NOAA20 and OLCIB data will be incorporated into the GlobColour chain. Line 5: meters Line 11: Do you mean "The redundancy can decrease the level of uncertainty"? Line 16: could you provide more information on the reference of the CCI product Line 27: use them

Page6:

Lines 1-2: "The RRS merging approach is a very attractive solution. However the issues linked to the instrument and difficulties of calibration shows that is challenging to be successful with this approach." This assertion is not really proven. Line 7; requires

---

## Referee Comment (RC2) · Emmanuel Boss (Referee) · 12 Feb 2019

Reviewer: Emmanuel Boss, University of Maine This paper describes the chlorophyll products based on Satellite Observation and disseminated in the frame of the Copernicus Marine Environmental Monitoring Service (CMEMS). Different strategies for merging remote sensing data are presented (e.g. merging radiance vs. merging the products) and the choice of using a merged product approach is justified.

While I see the benefit in publishing this paper, in its current form, it reads like a report to a funding agency rather than a scientific paper. In addition, this paper can benefit a lot from being read by a native English speaker (I am not).

For example, I would have expected that the comparison between products will also

involve the statistics of distributions of values of chlorophyll (histograms) as done when such algorithms are published (both globally and regionally). Statements are made in the conclusion sections that are not justified by results. Figures look like they were taken out of a report or a powerpoint presentation rather than high quality publication type figures (with cryptic titles, which is not the norm in papers, lines that don't show as in Fig. 4). Some results are provided in an external private website (http://hermes.acri.fr/index.php?class=animation). It they are to be considered they should be in the public domain.

I am returning an annotated PDF with many detailed comments.

Please also note the supplement to this comment:
https://www.ocean-sci-discuss.net/os-2018-155/os-2018-155-RC2-supplement.pdf

**Supplement:**

[revised manuscript text omitted]

---

## Author Comment (AC1) · 5 Apr 2019

Please see the figure and the supplemnt attached.

Please also note the supplement to this comment:
https://www.ocean-sci-discuss.net/os-2018-155/os-2018-155-AC1-supplement.pdf

[Figure]

Dear Referee,

Please find our comments/responses in blue in your text.
We have also attached a new release of the article. This a major revision of the initial article:
the form has been changed, the assertion are better argued. Most of the figures have been also reviewed.

Thank you for your useful comments.

Philippe Garnesson on behalf of co-authors.

Anonymous Referee #1

This article presents the new version of the GlobColour product delivered by ACRIST within the CMEMS. As this GlobColour Chlorophyll-a (Chl-a) product has a global coverage and provides retrievals in coastal waters this manuscript can be of interest for many current and future users of satellite-derived products.

Chl-a in this new GlobColour product is derived from two algorithms: the Color Index of NASA for clear waters (Chl-a < 0.15 mg m-3) and the OC5 algorithm of Ifremer for water where Chl-a is superior to 0.2 mg m-3, including the coastal turbid waters. This is very similar to the strategy chosen by the Plymouth Marine Laboratory for the OC-CCI product, also provided at global scale. However a distinction is clearly made by the authors: the GlobColour processing chain provides a Level 3 Chl-a multi-sensor product obtained from mono-sensor Chl-a whilst the Level-3 Chl-a of OC-CCI is obtained from C1 OSD Interactive comment Printer-friendly version Discussion paper a prior merging of the remote-sensing reflectance of the different sensors on a common reference of spectral bands (SeaWiFS).

The OC-CCI approach is similar to that of the Mediterranean Product Unit of CMEMS described in Volpe et al. (Ocean science, accepted). Targeting directly Chl-a, the GlobColour processing can theoretically and practically be adapted more quickly to the modification of the products of any single sensor (following the reprocessing by the Agencies) whilst this task is more difficult to achieve through the complexity of the band switch and band correction operated in the OC-CCI approach. However as pointed out by Volpe et al., the band merging approach has the advantage of providing a homogeneous dataset of spectral reflectance from which can be derived, in full consistency for the long term, different environmental parameters, amongst them Chl-a but also light attenuation, Kd, Suspended Particulate Matter, . . ..

Yes, we fully agree with Volpe, and "line 16, section 2.1", it was indicated <<the approach is theoretically very attractive>> but the promising consistency supposed the input reflectances are consistent. We add a paragraph to discuss advantages and drawbacks underlined by Volpe in part 3.1 (p 4, lines 19-23).

The authors discuss different issues encountered in the near real-time and long term processing of Ocean Colour data and some interesting illustrations are provided on the effect of the drift of Rrs in flight and the

**Fig. 1.**

**Supplement:**

[revised manuscript text omitted]

---

## Author Comment (AC2) · 5 Apr 2019

Please see the figure and the document attached.

Please also note the supplement to this comment:
https://www.ocean-sci-discuss.net/os-2018-155/os-2018-155-AC2-supplement.pdf
* * *
Dear Emmanuel Boss,

Please find our comments/responses in blue in your text.
We have also attached a new release of the article. This a major revision of the initial article:
the form has been changed, the assertions are better argued. Most of the figures have been also reviewed.
All your comments have been taken into account but there were so much (thank you for your efforts) that
it is difficult to track them.
If required we can also provide a document with the word track change (but I am afraid it will be not very
useful).

We have attached a new release

Thank you for your useful comments.

Philippe Garnesson on behalf of co-authors.

Emmanuel Boss (Referee) emmanuel.boss@maine.edu

Reviewer: Emmanuel Boss, University of Maine

This paper describes the chlorophyll products based on Satellite Observation and disseminated in the
frame of the Copernicus Marine Environmental Monitoring Service (CMEMS). Different strategies for
merging remote sensing data are presented (e.g. merging radiance vs. merging the products) and the
choice of using a merged product approach is justified.

While I see the benefit in publishing this paper, in its current form, it reads like a report to a funding
agency rather than a scientific paper. In addition, this paper can benefit a lot from being read by a native
English speaker (I am not).

The form of the paper has been reviewed.

For example, I would have expected that the comparison between products will also involve the statistics
of distributions of values of chlorophyll (histograms) as done when such algorithms are published (both
globally and regionally).

The objective of this article is to highlight two major topics (merging and flagging) and to justify the
approach selected for the GlobColour CMEMS processor (there is no innovation proposed in terms of
chlorophyll algorithm retrieval). We have modified title/abstract/conclusion to better explain our
objective.

Statements are made in the conclusion sections that are not justified by results.

The conclusion has been fully reviewed.

**Fig. 1.**

---

## Referee Report (RR1)

Dear Emmanuel Boss and anonymous Referees,

Please find below 4 documents:

- 1. The authors comments/responses (in blue) inserted in the review done by Emmanuel Boss.
- 2. The authors comments/responses (in blue) inserted in the review done by the anonymous Referee.
- 3. A new release of the article. This a major revision: the form has been changed and the assertions are better argued. Most of the figures have been also reviewed. The note https://www.ocean-sci-discuss.net/os-2018-155/os-2018-155-RC2-supplement.pdf has been also taken into account.
- 4. The same release of the article but with the word track changes.

Thank you for again for your useful comments.

Philippe Garnesson on behalf of co-authors.

**1/ Answers (in blue) in the note (in black) of Emmanuel Boss**

Emmanuel Boss (Referee) emmanuel.boss@maine.edu

Received and published: 12 February 2019

Reviewer: Emmanuel Boss, University of Maine

This paper describes the chlorophyll products based on Satellite Observation and disseminated in the frame of the Copernicus Marine Environmental Monitoring Service (CMEMS). Different strategies for merging remote sensing data are presented (e.g. merging radiance vs. merging the products) and the choice of using a merged product approach is justified.

While I see the benefit in publishing this paper, in its current form, it reads like a report to a funding agency rather than a scientific paper. In addition, this paper can benefit a lot from being read by a native English speaker (I am not).

**The form of the paper has been reviewed.**

For example, I would have expected that the comparison between products will also involve the statistics of distributions of values of chlorophyll (histograms) as done when such algorithms are published (both globally and regionally).

The objective of this article is to highlight two major topics (merging and flagging) and to justify the approach selected for the GlobColour CMEMS processor (there is no innovation proposed in terms of chlorophyll algorithm retrieval). We have modified title/abstract/conclusion to better explain our objective.

Statements are made in the conclusion sections that are not justified by results.

**The conclusion has been fully reviewed.**

Figures look like they were taken out of a report or a powerpoint presentation rather than high quality publication type figures (with cryptic titles, which is not the norm in papers, lines that don't show as in Fig. 4).

**All figures have been revisited.**

Some results are provided in an external private website (http://hermes.acri.fr/index.php?class=animation). It they are to be considered they should be in the public domain.

**The Reference to the website has been removed (out of scope).**

I am returning an annotated PDF with many detailed comments.

All these comments were fully pertinent and have been fixed: form, redundancy, statements justified (or removed).

Please also note the supplement to this comment: https://www.ocean-sci-discuss.net/os-2018-155/os- 2018-155-RC2-supplement.pdf

**2/ Answer (in blue) in the note (in black) of the anonymous Referee.**

Anonymous Referee #1

Received and published: 24 January 2019

This article presents the new version of the GlobColour product delivered by ACRIST within the CMEMS. As this GlobColour Chlorophyll-a (Chl-a) product has a global coverage and provides retrievals in coastal waters this manuscript can be of interest for many current and future users of satellite-derived products.

Chl-a in this new GlobColour product is derived from two algorithms: the Color Index of NASA for clear waters (Chl-a < 0.15 mg m-3) and the OC5 algorithm of Ifremer for water where Chl-a is superior to 0.2 mg m-3, including the coastal turbid waters. This is very similar to the strategy chosen by the Plymouth Marine Laboratory for the OC-CCI product, also provided at global scale. However a distinction is clearly made by the authors: the GlobColour processing chain provides a Level 3 Chl-a multi-sensor product obtained from mono-sensor Chl-a whilst the Level-3 Chl-a of OC-CCI is obtained from C1 OSD Interactive comment Printer-friendly version Discussion paper a prior merging of the remote-sensing reflectance of the different sensors on a common reference of spectral bands (SeaWiFS).

The OC-CCI approach is similar to that of the Mediterranean Product Unit of CMEMS described in Volpe et al. (Ocean science, accepted). Targeting directly ChI-a, the GlobColour processing can theoretically and practically be adapted more quickly to the modification of the products of any single sensor (following the reprocessing by the Agencies) whilst this task is more difficult to achieve through the complexity of the band switch and band correction operated in the OC-CCI approach. However as pointed out by Volpe et al., the band merging approach has the advantage of providing a homogeneous dataset of spectral reflectance from which can be derived, in full consistency for the long term, different environmental parameters, amongst them ChI-a but also light attenuation, Kd, Suspended Particulate Matter, . . ..

Yes, we fully agree with Volpe, and "line 16, section 2.1", it was indicated << the approach is theoretically very attractive>> but the promising consistency supposed the input reflectances are consistent. We add a paragraph to discuss advantages and drawbacks underlined by Volpe in part 3.1 (p 4, lines 19-23).

The authors discuss different issues encountered in the near real-time and long term processing of Ocean Colour data and some interesting illustrations are provided on the effect of the drift of Rrs in flight and the successive reprocessings by the Agencies onto the OC-CCI product (Fig. 2 & 3) or onto the GSM/Nasa (Fig. 5.) product.

However these major operational constraints have also consequences on the GlobColour and these latters are not described Thorough descriptions of the quality and flaws of the GlobColour products over the period 1997-2018 are missing.

Because previous remark about Volpe, it is important to highlight that the input reflectances (merged or not merged) have issues with consistency. Clearly GlobColour is impacted in the same way as OC-CCI (and it was indicated line 30 of section 3.2). We have added the figure 7 and comments to be clearer on this point.

This could be due to the fact that this GlobColour product is new. In that case, would it not be better to change the title for something as "The new GlobColour product and the challenge of Ocean Colour processing at global scale?" The choices of the author in the article should be easier to understand.

The objective of this article is to highlight two major topics (merging and flagging) and to justify the approach selected for the GlobColour CMEMS processor. So, we agree the title should be more explicit. We have changed it with "The CMEMS GlobColour Chlorophyll-a Product Based on Satellite Observation: multi-sensor merging and flagging strategies."

In conclusion of these general remarks, the paper should be re-organized. For instance, the first paragraph of the Result and Discussion, 3.1, deals only with the OC-CCI and GSM products!

GSM discussion and plots have been removed: it was just to illustrate the sensor drifts, but we agree it was confusing to introduce another algorithm.

Finally, in this text, the GlobColour product is assessed through its difference (more flexibility as for ingesting OLCI data) with other products but not through comparisons to in situ data; which is a main issue in OC applications, particularly in the coastal waters that are now clearly addressed by OC-CCI and GlobColour. This issue should, at minimum, appear in the discussion.

As CMEMS OC products aim at covering coastal waters, why not take advantage of the coastal monitoring networks for a better flexibility in modifying the processing chains after a reprocessing by the Agencies or the availability of a new sensor?

As said previously, the objective of this paper is to highlight strategies used by CMEMS GlobColour for merging and flagging. It will be of course pertinent to be able to justify strategy using an in situ assessment. However, from our point of view this is a utopic objective: number of in situ are too limited especially on coastal area, the quality of the in situ and the satellite observation are highly questionable (e.g. change of 10% of chlorophyll with NASA-R2018, drifts of sensors). Of course, for CMEMS we are providing a global assessment, but badly it cannot be used to justify merging or flagging approach.

Despite these comments, the manuscript is worth publishing. The illustrations are very informative and the main issues in the operational processing of OC data are pointed out. Specific remarks Introduction Line 11 the CCI/Sv3 is mentioned but not referenced.

**It is now referenced at the first citation in the abstract.**

That will be done later in the text. Line 19 The continuity between OCx (OC3 & OC4) and OC5 is guaranteed by the construction of the OC5 tables. What do you mean, OC3 and OC4 are used in the GlobColour product in complement to CI and OC5?

**In fact, OC5 is using a lookup table based on the ratio used by OC3/OC4. This table is only adjusted for complex water. The text about this topic has been fully reviewed (section 2.1 lines 24-26)**

2.2 Flagging approach This chapter is not clear. How is the OC5 LUT doing its own flagging? Not sure there is a control of the errors coming from the atmospheric correction or clouds by the ratios used in OC5. The

412 Rrs processed in OC5 can take into account a possible effect of the overcorrection of the atmospheric content but it could only be marginal in case or clouds, . . ..

The level2 upstream products of agencies are provided with flags and some official recommendation to apply them. For instance, if one of the reflectance associated to a pixel is negative this is suspicious, and it can be better to not use this pixel, however in practice other reflectance can be valid. All these flags are based on threshold set up to have the "best" compromise between quality and coverage. However, sometimes the official flags are not working (e.g.specially to determine the frontier of clouds). OC5 flagging is based on the flags from agency and empirical test which permits to improve the coverage (e g.the sun zenith angle (SZA) is set to 78° instead 70°). The example talking about "atmospheric conditions" was misleading and has been removed.

3.1 Results are those of OC-CCI (already mentioned). However this chapter deals with interesting issues in OC monitoring. In Figure 6, it would be better to show the 2 deviations with a same reference: MODIS Rrs 667. OLCI-MODIS and VIIRS-MODIS would appear with similar colours, demonstrating the variability of the atmospheric corrections between OLCI and VIIRS on one side and MODIS on the other.

**It has been changed (figure 5)**

Conclusion It is not really a conclusion. The conclusion (or the last lines of the discussion) should open a window on a possible improvement of the processing chains; I would have appreciated a larger view, a strategy for improving the quality of the products, an opening to validation and flexibility after the launch of a new sensor or after a major reprocessing by the Agencies.

**The conclusion has been fully rewritten.**

Technical corrections Page 1 (fixed or justified/removed in the new submission): Line 12: provides Line 24: the Ocean Colour Thematic Assembly Centre Page 2: Line 12; the CCI/S3v project is not defined Lines 16-19; the continuity of the 3 algorithms. CI , OC5 and ? Lines 20-21: the two sentences could be merged. Page 3: Lines 2 and 3. you already said (page 2 line 29) that VIIRS-NOAA20 and OLCIB data will be incorporated into the GlobColour chain. Line 5: meters Line 11: Do you mean "The redundancy can decrease the level of uncertainty"? The idea expressed Line 16: could you provide more information on the reference of the CCI product Line 27: use them Page6: Lines 1-2: "The RRS merging approach is a very attractive solution. However, the issues linked to the instrument and difficulties of calibration shows that is challenging to be successful with this approach." This assertion is not really proven. Line 7; requires

**The CMEMS GlobColour Chlorophyll-a Product Based on Satellite Observation: multi-sensor merging and flagging strategies.**

Philippe Garnesson1, Antoine Mangin1, Odile Fanton d'Andon1, Julien Demaria1, Marine Bretagnon1

1ACRI-ST, Sophia-Antipolis, 06904, France

[revised manuscript text omitted]

---

## Author Response (AR2)

Dear Referees,

All your remarks about the English and about the presentation have been considered. The document has been reviewed by a native speaker. Sorry to not take the time to do that in the previous releases.

About the 2 remarks :

- The flagging approach : The idea is not to appropriate a work on flagging that has been done by IFREMER (F. Gohin). The flagging is part of the source code that was provided by IFREMER. However, as suggested, it is good to clarify our positioning (section 2.2 has been modified in this way).
- The algorithms used: It seems the section 2.1 with the information about the new release was misleading. I confirm we are using the 3 algorithms (taking benefit of CI since Nov 2018). So, we have slightly modified the introduction to better introduce the section 2.1 where it is explained that when using OC5, we are in fact using mainly OCX which has been adapted for some range of values.

Please find below the new release with and without track changes. The improvement about figures is not part of the track change, but all have been revisited.

Thank you again for your contributions.

Philippe Garnesson on behalf the authors.

[revised manuscript text omitted]